# Universal Microcarriers Based on Natural and Synthetic Polymers for Co-Delivery of Hydrophilic and Hydrophobic Compounds

**DOI:** 10.3390/polym14050931

**Published:** 2022-02-25

**Authors:** Olga Yu. Kochetkova, Tatiana S. Demina, Olga Yu. Antonova

**Affiliations:** 1Institute of Theoretical and Experimental Biophysics Russian Academy of Science, Institutskaya Str. 3, 142290 Puschino, Russia; olga.antonova.iteb@gmail.com; 2Enikolopov Institute of Synthetic Polymeric Materials, Russian Academy of Sciences, Profsouznaya Str. 70, 117393 Moscow, Russia; detans@gmail.com

**Keywords:** layer-by-layer, microcapsules, polymers, hydrophilic and hydrophobic compounds, drug delivery

## Abstract

Several variants of hybrid polyelectrolyte microcapsules (hPEMC) were designed and produced by modifying in situ gelation methods and layer-by-layer (LbL) techniques. All of the hPEMC designs tested in the study demonstrated high efficiency of the model hydrophilic compound loading into the carrier cavity. In addition, the microcarriers were characterized by high efficiency of incorporating the model hydrophobic compound rhodamine B isothiocyanate (RBITC) into the hydrophobic layer consisting of poly-(d,l)-lactide-co-glycolide (PLGA), oligo-(l)-lactide (OLL), oligo-(d)-lactide (OLD) and chitosan/gelatin/poly-l-lactide copolymer (CGP). The obtained microcapsules exhibited high storage stability regardless of the composition and thickness of the polyelectrolyte shell. Study of the impact of hybrid polyelectrolyte microcapsules on viability of the adhesive L929 and suspension HL-60 cell lines revealed no apparent toxic effects of hPEMC of different architecture on live cells. Interaction of hPEMC with peritoneal macrophages for the course of 48 h resulted in partial deformation and degradation of microcapsules accompanied by release of the content of their hydrophilic (BSA–fluorescein isothiocyanate conjugate (BSA-FITC)) and hydrophobic (RBITC) layer. Our results demonstrate the functional efficiency of novel hybrid microcarriers and their potential for joint delivery of drugs with different physico-chemical properties in complex therapy.

## 1. Introduction

Most of the current systems of drug delivery enable loading and delivery of a single therapeutic agent [1,2,3,4]. However, there is a big number of diseases which ideally require complex treatment, e.g., different types of cancer or chronic inflammatory disorders. Simultaneous delivery of several compounds with different physical and chemical properties into the same target cells/tissue is known as combination drug delivery. Combination drug delivery systems offer a number of advantages, such as alleviation of multiple drug resistance, pharmacological synergy between the delivered compounds, as well as options to regulate bioavailability of the drugs in order to minimize their side effects. A number of carriers for combination delivery have been developed in recent years based on nanoparticles, dendrimers, microcapsules, etc. The existing systems for hydrophobic compound delivery, emulsion-based and spraying drying techniques in particular, do not meet the requirements for wider translational application. These methods are limited by low efficiency of hydrophobic compound loading, high manufacturing costs, and above all, the complexity of synthesis. The method of precipitation on the template appears advantageous, since it allows to alleviate these shortcomings [5,6,7,8,9,10].

Polyelectrolyte microcarriers are particularly promising for combined delivery of hydrophilic and hydrophobic compounds because of the opportunity to utilize the different properties of various parts of the carrier. For instance, hydrophilic molecules (e.g., proteins) can be encapsulated into the internal part of the carrier, while hydrophobic molecules can be loaded into its multilayered shell [11]. Microspheres can be produced both from natural (chitosan, alginate, gelatin, dextran, poly-l-arginine) and synthetic polymers (PLGA, polystyrene sulfonate, polyallylamine hydrochloride,). A group of synthetic polymers widely used for drug delivery is represented by poly(α-hydroxy esters), including poly(lactic acid) (PLA), poly(glycolic acid) (PGA), and their copolymers (PLGA) [12,13]. These polymers have been approved by FDA due to their high biocompatibility, low immunogenicity, and controlled biodegradation rates. However, carriers based on these polymers often demonstrate relatively low loading capacity [14].

The possibility to produce natural/synthetic composites of hybrid nature enables to combine the advantages inherent to each of these components and increase the loading capacity, while retaining biocompatibility and biodegradation capacity. By varying the conditions of synthesis it is possible to control the speed of scaffold degradation and its physico-chemical properties (solubility, pH sensitivity). In particular, copolymers of chitosan, gelatin, and poly(lactic acid) (CGP) are relevant biodegradable carriers of hydrophobic nature which allows delivery of hydrophobic drugs with a high loading capacity [15,16,17,18,19,20]. One of the notable representatives of natural polymers is chitosan, which currently has broad application in production of biomaterials and various drug delivery systems [15,16]. Chitosan possesses a number of advantages, such as biocompatibility, antimicrobial and antimycotic activity. The presence of functional groups (predominantly positively charged) in the chitosan molecule is particularly useful, since it enables relatively easy chemical modification for various biomedical applications [16,17].

Wang et al. [8] presented a novel magnetically sensitive hydrophilic-hydrophobic system with controlled loading and release of Ibuprofen (IBU) and doxorubicin (DOX). IBU in this system demonstrated faster release compared to DOX The authors also demonstrated that the release of both drugs was decelerated with the increase of the number of polyelectrolyte layers. Luo et al. [14] introduced a similar system in which IBU was incorporated into the hydrophobic PLGA layer, while the hydrophilic core was loaded with Fe_3_O_4_ nanoparticles. The efficiency of IBU encapsulation was similar to that reported by Wang et al. [8]

Here we describe fabrication and properties of hybrid microcapsules which utilize copolymers of chitosan, gelatin, and either poly(lactic acid) or low molecular weight isolactides as carriers for hydrophobic compound delivery. We chose the popular PLGA carrier to serve as the control standard of polymer for hydrophobic layer synthesis in our study. Hybrid microcapsules were produced by a modified method of sequential adsorption of oppositely charged polyelectrolytes on the surface of layer-by-layer (LbL) carbonate matrix with modifications [14]. We used BSA-FITC as the model hydrophilic compound for encapsulation within the microspheres and RBITC for loading into the hydrophobic layer. We characterize the ultrastructure of the microcapsules and analyze the differences of loading efficiency and release kinetics of the model compounds between microcapsules of different design; we also demonstrate internalization of the microcarriers and intracellular release of the cargo.

## 2. Materials and Methods

Poly(DL-lactide-co-glycolide) 50:50 (PLGA, Mw 45 kDa) was obtained from Boehringer Ingelheim (Belgium); oligo-(l)-lactide (OLL, Mw 5 kDa), oligo-(D)-lactide (OLD, Mw 5 kDa) were synthesized from the respective lactic acid isomers as described earlier [14]; Dextran sulphate sodium salt (DS) (Mw~9–15 kDa), Poly-l-arginine hydrochloride (PAr, Mw~15–70 kDa), Polyallylamine hydrochloride (PAH, Mw~58 kDa), Poly(sodium 4-styrenesulfonate) (PSS) (Mw~70 kDa), Poly vinyl alcohol (PVA, Mw 70 kDa), Fluorescein isothiocyanate (FITC), Rhodamine B isothiocyanate (RBITC), Resazurin sodium salt, Crystal Violet, EDTA, zimozan were all obtained from Sigma-Aldrich (Saint Louis, MO, USA); Phosphate buffered Saline (PBS), DMEM/F12, trypsin, penicillin and streptomycin were purchased from PanEco (Moscow, Russia); Fetal calf serum (FCS) was obtained from Invitrogen (Carlsbad, CA, USA); Chloroform was obtained from Chimmed (Moscow, Russia); DMSO was obtained from Panreac (Castellar del Vallès, Spain).

### 2.1. Hybrid PEMC Synthesis

The CaCO_3_ microparticles used as templates for the production of microcapsules were prepared according to the previously published method [21]. Briefly, equal volumes of 1 M aqueous solutions of CaCl_2_ and Na_2_CO_3_ were mixed at 350 rpm for 30 s by stirring; the suspension was left for 5–7 min until complete clarification of the supernatant. The resulting CaCO_3_ particles were centrifuged for 30 s at 4000× *g*, washed three times with twice distilled water and dried.

Chitosan/gelatin/poly-l-lactide copolymer (CGP) was synthetized via mechanochemical approach by treatment of solid powder mixtures of chitosan (Mw 60 kDa, DA 0.1), poly(l)-lactide (Mw 160 kDa) and gelatin (chitosan: PLLA: gelatin ratio as 52:35:13 wt.%) in a Berstorff ZE-40 twin-screw extruder (Krauss Maffei Berstorff, Munich, Germany) [22], after which it was reconsituted in chlorophormPLGA, OLL and OLD, were dissolved in DMSO.

The method for hybrid polyelectrolyte microcapsule production was adapted from [14]. Fluorescent probe RBITC was used as the model hydrophobic compound, 1 mL of RBITC solution (70 mg/mL in DMSO was added to 1 mL of 1% PLGA (OLL, OLD, CGP) solution. The resulting mixture was incubated with 10 mg of CaCO_3_ microparticles (or composite microparticles CaCO_3_ with BSA-FITC), prepared as described above, for ~1 h with constant stirring. Subsequently, the sediment was collected by centrifugation at 4000× *g* for 1 min and washed once with DMSO (chloroform in the case CGP). The resulting templates were immediately mixed with 1 mL 0.2% PVA aqueous solution by vortexing for 3–4 min. The resulting CaCO_3_ microparticles with the hydrophobic layer coating were collected by centrifugation at 4000× *g* for 1 min and then washed with water; the centrifugation and washing steps repeated twice. These microparticles were then used as a template which was coated with biodegradable (PAr/DS) or non-biodegradable polymers (PAH/PSS) using LbL techniques, as previously described [4]. Briefly, PAr/DS and PAH/PSS were dissolved in 0.5 NaCl (2 mg/mL). Each polyelectrolyte layer was formed by application of the corresponding polymer solution for 15 min, after which capsules were centrifuged and washed three times with 0.5 M NaCl. 1–3 s ultrasound pulses were used to break particle aggregates. After that, the CaCO_3_ templates were removed by incubation in 0.2 M EDTA, pH 7.4 for 40 min followed by washing with water. After that the resulting hPEMC were reconstituted again in 1 mL of 0.2 M EDTA, pH 7.4 and left overnight, followed by another washing step with water. At the final step hPEMC were separated from the supernatant by centrifugation at 6000× *g* for 5 min.

#### 2.1.1. Loading of FITC-BSA, RBITC into the Hybrid Microcapsules

Encapsulation efficiency was determined by measuring the model compound (*MC*) content in the supernatant after hPEMC synthesis. The content of the encapsulated material ($Q_{MC_{enc}}$) was calculated using the formula:$$Q_{MC_{enc}}=\frac{Q_{MC_{1}}-Q_{MC_{s}}\;}{Q_{MC_{1}}},$$

Encapsulation efficiency (*Enc*, %) was defined as:$$Enc=\frac{Q_{MC_{1}}-Q_{MC_{s}}\;}{Q_{MC_{1}}}\times 100\%$$
where $Q_{MC_{1}}$ is the initial amount of *MC* in the aliquot used for encapsulation, $Q_{MC_{s}}$ is the amount of *MC* in the supernatant of the sample. The *MC* content of samples was determined fluorometrically using an Infinite F200 (Tecan, Switzerland).

#### 2.1.2. Drug Release from the Hybrid Microcapsules

The release behaviors of model compounds from the hybrid microcapsules were studied in 2 mL of phosphate buffered saline (PBS; pH 7.2) at 37 °C under constant shaking. After defined time intervals, the samples were centrifuged, and the supernatant was collected and replaced by fresh PBS. BSA-FITC and RBITC release was determined using fluorescence spectrometer (Infinite 200, Tecan, Switzerland). All experiments were done in triplicate to ensure reproducibility.

The cumulative release (*Rel*, %) was calculated using the following equation:$$Rel=\frac{Q_{MC_{enc}}-Q_{MC_{t}}\;}{Q_{MC_{enc}}}\times 100\%$$
where $Q_{MC_{t}}\;i$s the amount of *MC* at the predetermined time points.

### 2.2. Characterization of hPEMC

#### 2.2.1. Zeta Potential Measurement

The ζ-potential of the microparticles was determined using Malvern Zeta-sizer (Malvern Zetasizer Nano, UK). The coated CaCO_3_ microparticles were suspended in dH_2_O (pH 6.5), and the reported zeta potential values are the averages of three consecutive measurements.

#### 2.2.2. Transmission Electron Microscopy

Fabrication of hPEMC loaded with metal nanoparticles: biomineral CaCO_3_ cores containing BSA were incubated in 1% PLGA/CGP/OLL/OLD polymer solution with metal nanoparticles for 1 h. Magnetic iron nanoparticles were produced as described in [23]. The resulting cores with an external hydrophobic layer were coated with (PAr/DS)_2_ after which the matrix was demineralized in EDTA solution as described above in 2.1. Microcapsules were fixed for 1 h in a 5% aqueous glutaraldehyde solution. The resulting suspensions were centrifuged for 5 min at 3000 rpm, and the pellets were resuspended for complete fixation in a 1% aqueous osmium tetroxide solution. After fixation for 1 h, the suspensions was centrifuged again, and the pellets were dehydrated by successive washing with 30%, 50%, 75%, and 90% acetone in water and then with 100% anhydrous acetone, after which they were soaked in acetone–epoxide resin Epon 812 at the 1:1 and 1:3 ratios for 12 h in each mixture, and embedded into fresh resin. Upon completion of resin polymerization, 70 µm sections of the samples were prepared by a conventional method using LKB3 ultratome (Sweden) with glass knives. Sections were trapped onto supporting nets, contrasted with an aqueous solution of uranyl acetate (1.5 h) and lead citrate (30 min), and examined in a Tesla BS500 electron microscope (Brno, Czech Republic)) at an accelerating voltage of 90 kV and a magnification of ×18,000 [24]. The average wall thickness of each type of PEMC was calculated based on the measurements performed for 50 microcapsules.

### 2.3. Cell Culture Experiments

L929 murine fibroblast and HL-60 human promonocytic cell lines were purchased from Russian Cell Culture Collection (RCCC, Institute of Cytology of RAS) and maintained in 25 cm^2^ culture flasks in DMEM/F-12 culture medium supplemented with 10% (*v*/*v*) fetal bovine serum and antibiotics (100 mg/mL penicillin–streptomycin). Cells were incubated at 37 °C in an atmosphere of 5% CO_2_ and 95% air with more than 95% humidity; medium was changed every two days for cell feeding. 

#### 2.3.1. In Vitro Cytotoxicity Assay

L929 and HL-60 cell lines were seeded in 96-well plates at the density of 5 × 10^3^ cells per well. hPEMC were added to the cells at 2:1 and 10:1 ratios (hPEMC: cells) 24 h after seeding with 0.5 µg/mL PLGA, CGP, OLL, OLD and incubated for 72 h. Wells containing the same cells without hPEMC and polymers served as a control. 

Viability of HL-60 cells was determined by resazurin assay. Cells were incubated with 32 µg/mL resazurin sodium salt for 4 h at 37 °C, 5% CO_2_. Fluorescence intensity at 595 nm was measured using Infinite F200 fluorescence microplate reader Tecan (Zürich, Switzerland).

Viability of L929 cells was determined by Crystal Violet assay. Cells were fixed in 70% ethanol for 30 min and then incubated in 0.5% Crystal Violet dye aqueous solution for 15 min. After the incubation was completed, 100 µL of 1% SDS solution was added to each well and optical density was measured at 540 nm using Infinite F200 Fluorescence Microplate Reader. 

Viability was estimated by comparing the numbers of live cells in the experimental wells with hPEMC/polymer with the live cell numbers in the control wells.

#### 2.3.2. Hybrid Polyelectrolyte Microcapsule Interactions with Rat Macrophages

Animal studies were performed on male Wistar rats with body mass 200–250 g. For peritoneal macrophage isolation rats were injected intraperitoneally with 50 mg/kg zymozane solution and sacrificed 24 h post injection. Macrophages were isolated by peritoneal lavage as described in [25]. The resulting cell suspension was centrifuged at 300× *g* for 4 min and the cell pellet was resuspended in DMEM/F12 medium with 10% FCS. 2 × 10^5^ cells in 2 mL of medium were seeded on glass coverslips placed in cell culture Petri dishes. 24 h after seeding cells were treated at a 1:2 capsule to cell ratio with hPEMC containing BSA-FITC encapsulated in the microcapsule interior and RBITC in the hydrophobic external layer formed by PLGA, CGP, OLL or OLD. After incubation with the microcapsules for 48, 72 or 96 h cells were fixed in 4% paraformaldehyde solution for 12 h at 4 °C and prepared for confocal microscopy imaging (Leica TCS SP5, Wetzlar, Germany.).

### 2.4. Statistical Analyses

Data analysis and visualization was performed with Origin Pro (v. 2019b 9.6.5.169). The results were presented as mean ± standard error of mean. The statistical significance of observed differences was determined by Student’s *t* test. The results were considered statistically significant for *p*-values < 0.05.

## 3. Results

By combining the methods of in situ gelation of PLGA/CGP/OLL/OLD and layer-by-layer adsorption of oppositely charged polyelectrolytes we have developed hybrid LbL microcapsules for efficient loading and delivery of hydrophobic and hydrophilic compounds. For comparison, we used PEMC capsules composed of PSS/PAH as prototype synthetic nondegradable capsules, and of PAr/DS—as prototype biodegradable capsules. (Figure 1).

### 3.1. Determination of the ζ-Potential of hPEMC

The measurements of the ζ-potential of CaCO_3_ particles in aqueous solution have yielded values around –15.8 mV (Figure 2). Coating of spherulites with a hydrophobic layer of the chosen polymers was performed during incubation of dry templates in organic solvents (DMSO, chloroform); the polar molecules of these solvents were shielding the surface charges on the microparticles, shifting their ζ-potential towards neutral values. The formation of polymer coating changed the ζ-potential of the particles to −2.4 mV for PLGA, −1.26 mV for OLL, and −2.5 mV for OLD. The particles coated with CGP had the highest negative ζ-potential of −9 mV. 

### 3.2. Ultrastructural Characterization of Biodegradable hPEMC

Figure 3A,B demonstrates a SEM image of CaCO_3_ particles and CGP(PAr/DS) hPEMC after drying. The average size of carbonate particles was 3–5 µm. The surface of the microspheres displays a rough and friable texture and they appear slightly deformed, but retain the overall spherical shape, which demonstrates their mechanical stability.

Surface deposition of PLGA/CGP/OLL/OLD on the CaCO_3_ matrix and microcapsule integrity after removal of the carbonate matrix were confirmed by TEM (Figure 3) as described in Methods, Section 2.2.2. In order to characterize the ultrastructure of hybrid microcapsules by TEM we prepared hPEMC containing iron oxide nanoparticles. As seen in Figure 3C,D, the polymer layer of PLGA/CGP/OLL/OLD (PAr/DS)_2_ microcapsules shows diffuse staining and there is no contrast between the internal and external parts of the particles, which makes us suggest that the resulting hPEMC have uniform coating. The thickness of the layer was 19.7 ± 1 nm for PLGA, 17.5 ± 5.3 nm for CGP, 12.8 ± 4 nm for OLL, 12.8 ± 4 for OLD (Figure 3E), as determined by image analysis using ImageJ.

It should be mentioned that, although incorporation of magnetic particles into hPEMC was not our main goal and was used as a supplementary technique for structural characterization, production of microcapsules with this type of loading may have its own practical applications. In particular, it offers additional means for targeted delivery of the containers and controlled drug release [8].

### 3.3. Efficiency of Model Compound Encapsulation

All variants of hPEMC design used in the study demonstrated high efficiency of model compound (BSA-FITC) incorporation into the carrier cavity (Table 1). The efficiency of incorporation of the hydrophobic model compound (RBITC) into the hydrophobic layer formed by the polymers PLGA, CGP, OLL and OLD around the particle core was also high (at least 77% for PLGA and 91% in the case of CGP).

### 3.4. Analysis of the Release Profiles of the Model Hydrophilic and Hydrophobic Compounds

To study the impact of capsule design and the thickness of polyelectrolyte coating upon release of encapsulated BSA-FITC and RBITC we analyzed the content release profiles during incubation of PEMC in PBS at 4 °C and 37 °C (Table 2).

The results of the analysis demonstrate high levels of hPEMC stability during long-term incubation both at 4 °C and 37 °C, regardless of the microcapsule composition (which also affects the thickness of the polyelectrolyte shell and the layer of polymer used for loading hydrophobic drugs). Incubation of microcapsules containing Polymer(PAH/PSS) and Polymer (PAH/PSS)_2_ at 4 °C resulted in low release of BSA-FITC and RBITC; for instance, OLL (PAH/PSS) microcapsules released only 1.3% of the total BSA-FITC content of the capsules retained after microspherulite removal.

### 3.5. Interaction of Hybrid PEMC with Cultured Cells and Analysis of Potential Cytotoxicity

Cytotoxic effects of the hPEMC developed in the study were tested against two different cell lines conventionally used for toxicological studies; L929 mouse fibroblasts were chosen as a model for adherent cells, while human promyelocytic cell line HL-60 was used as a model of cells growing in suspension. The effects of hPEMC of different architecture, and the corresponding blank polymers used for their synthesis on L929 mouse fibroblasts are summarized in Figure 4. Blank polymers PLGA, CGP, OLL, OLD applied at 0.5 µg/mL had no apparent toxic effect on the cells. Similarly, no apparent toxicity was detected in experiments with microparticles coated with PLGA, CGP, OLL, OLD, or with hPEMC containing a hydrophobic layer of PLGA/CGP/OLL/OLD and coated with PAr and DS biodegradable polymers; the results after 72 h of incubation were similar between experiments with capsule-to-cell ratios 2:1 and 10:1.

The safety of the novel microcarriers and the corresponding blank polymers PLGA, CGP, OLL, OLD was further confirmed in experiments with HL-60 cells (Figure 5). Isolated structural components of the microcapsules (PLGA, CGP, OLL, OLD) applied at 0.5 µg/mL had no effect on the viability of promyelocytes. Similarly, no toxicity was observed in the presence of template cores coated with PLGA, CGP, OLL, OLD or PLGA/CGP/OLL/OLD(PAr/DS) at the chosen hPEMC:cell ratios of 2:1 and 10:1.

### 3.6. Internalization of hPEMC by Rat Peritoneal Macrophages and Model Cargo Release

Next, we studied the interaction of the hPEMC produced by us with rat peritoneal macrophages. Macrophages are particularly important as a model for analyzing the functional properties of microcarriers and nanoparticles since they are potential therapeutic targets for the treatment of inflammatory disorders, atherosclerosis, and cancer [26].

The cellular uptake of hPEMC was analyzed by incubating them with cultured rat macrophages for 48 h. Figure 6 shows representative images obtained by confocal laser scanning microscopy. As seen from the confocal images, no changes in the morphology of peritoneal macrophages were observed; the cells retained the typical round shape. For each incubation time point, 3 independent experiments were performed, and 5 fields of view analyzed in each experiment. No significant morphological differences were observed between cells incubated with hPEMC of different composition. Exposure to microcapsules did not cause changes in the numbers of viable cells, and no dead or apoptotic cells could be detected; these results are in agreement with the toxicity tests reported above (Figure 4 and Figure 5), demonstrating non-cytotoxic nature of all of the materials used in this work: CaCO_3_ microparticles, the CGP composite polymer, OLL, OLD, and complete hPEMC themselves. Incubation for 48 h resulted in robust internalization of the microcapsules by macrophages, as seen in the confocal images. On average, 3 to 4 capsules were phagocytosed by each cell, in line with our previously published data with other cell models [4,27].

No significant differences in the number of internalized capsules per cell were observed between the different types of hPEMC used in the study. It is important to note that we have not observed any aggregated capsules, or capsules bound to the cell membrane without internalization, which demonstrates good quality of the microcapsule preparations.

Internalized microcapsules retained their initial size (3–5 µm). However, they demonstrated signs of partial degradation and cargo release, both from the hydrophobic layer (RBITC) and the hPEMC internal space (BSA-FITC), at the time of observation. 

## 4. Discussion

Co-precipitation remains one of the most popular techniques for drug molecule encapsulation inside the carrier. As it has been demonstrated previously, the method of co-precipitation allows to encapsulate (load) up to 80% of the protein present in the solution into the cavities of microcapsules during biomineral core formation [28]. Its major drawback, however, is significant loss of drug content during solubilization of the carbonate core particles (remineralization); this might be due to low retention of proteins by the polyelectrolyte shell. About 30% of protein load can be lost during core removal. The pore structure and the thickness of the polymer shell determine the speed of encapsulated compound release. Our results suggest that the presence of an additional hydrophobic layer composed of PLGA/CGP/OLL or OLD may be preventing free diffusion of BSA encapsulated in the microsphere cavity, and thus maintains prolonged release of the microsphere content; this feature may be particularly important for targeted delivery of therapeutic compounds.

The earlier study of microcapsules composed of (PAr/DS)_3_ reported a uniform shell thickness of 44.3 nm; reducing the number of PE layers resulted in impaired stability of the shell which disintegrated after removal of the carbonate core [29]. Our current results demonstrate that an additional hydrophobic layer facilitates the formation of a stable uniform microcapsule shell which requires a smaller number of polyelectrolyte (PE) layers. The parameters of thickness and mechanical strength of the PE shell are important for the development of drug delivery systems, and its enhanced stability increases the range of structures and compositions of PEMC which could be manufactured, depending on the particular drug properties and therapeutic applications.

One of the important parameters characterizing the colloidal stability of microparticles is the ζ-potential. Particles with the ζ-potential of +30 mV and lower, or −30 mV and higher, are considered colloidally stable. The surface potential of microparticles depends on a variety of factors, such as pH of the solution, ionic strength, etc. For instance, the ζ-potential of CaCO_3_ particles is positive in the presence of an excess of Ca_2_^+^ in the medium, but it is negative in the case when CO_3_^−^ ions are prevalent. Both positive and negative values of ζ-potential have been previously reported for CaCO_3_ particles [30]. Our results showed slightly negative values, which is consistent with the data obtained in the work [14]. The changes of the ζ-potential reflected effective coating of carbonate particles with the chosen polymers, while its low values indicated that the resulting particles had low colloidal stability. Subsequent stabilization of this formed layer was achieved by coating the microparticles with several additional layers of cationic and anionic polyelectrolytes using LbL deposition.

Our data on the encapsulation of a hydrophobic compound in a microcapsule shell are consistent with the publication of Luo et al. [14]; they loading efficiency of ibuprofen in their studies was ~25%. Similar results were reported in [8] for the model hydrophobic drugs ibuprofen and doxorubicin, with loading efficiencies of 19.4% and 32.25%, respectively. In our study (Results 3.3) the efficiency of incorporation of (RBITC) was significantly higher (77% for PLGA, 91% for CGP). It is not possible to directly compare these values due to differences in the physico-chemical properties of RBITC and the compounds mentioned above; however, RBITC as a model compound in our studies possessed an important experimental advantage, since it simultaneously served a fluorescent reporter, allowing for reliable quantitation and imaging of its distribution. It has been also used as a model hydrophobic molecule in other similar studies [31,32]. Our initial results are encouraging; however, further studies with hydrophobic drug molecules will allow to better determine the therapeutic potential of our microcapsule design. Data on RBITC incorporation into and release from the hydrophobic layer of the microcapsules demonstrate the potential advantages of oligolactides and CGP, as an alternative to the widely used PLGA, as efficient carriers for hydrophobic molecules. The thickness of the CGP layer in hPEMC was slightly (~2 nm) bigger, while the thickness of the oligolactide layer was ~5 nm smaller (due to lower molecular weight) than the typical thickness of the PLGA layer. These differences had negligible impact on loading efficiency which remained consistently high (80–90%).

Analysis of quantitative changes of BSA-FITC encapsulated in the process of container production has shown that the process of deposition of hydrophobic polymers on the surface of the carbonate core resulted only in minor release of protein content from the combinationspherulitesinto the solution. Significant loss of protein occurred during carbonate matrix removal with EDTA solution, as well as subsequent washes. These results are in agreement with the data from which demonstrated high efficiency of BSA incorporation into polyelectrolyte microcapsules [31].

Moreover, it should be noted that none of the differences in cumulative release between hPEMC with different polymer composition listed in the Table 2 were statistically significant. Other authors report robust release (up to 90%) of encapsulated hydrophobic and hydrophilic compounds within the first few hours of incubation which could be explained by the absence of additional layers of PAH and PSS coating on the microcapsules [8,14]. The presence of additional polymers in the shell prevented the burst release of encapsulated compounds, and this effect was proportional to the number of PAA and PSS layers [8,26].

Studies of the effect of PLGA, CGP, OLL, OLD polymers themselves, spherulites coated with them or hPEMC containing a hydrophobic layer of PLGA/CGP/OLL/OLD and coated with biodegradable PArg and DS polymers on cell lines L929 and HL-60 cells showed no toxic effect in all studied variants. This is consistent with the results obtained earlier by Demina et al. [22] where reported successful culturing of mouse fibroblasts using CGP films. The viability of fibroblasts cultured on CGP films was comparable to that of the fibroblasts cultured on chitosan films, while culturing on polylactide homopolymer resulted in cell viability not exceeding 30% of the control. Apparently, the presence of chitosan and gelatin in the CGP molecule could have a knock-on effect in the bulk of the films on their bioactivity.

Since hybrid microcapsules are expected to be used as a delivery system to mammalian cells and tissues, we studied their interaction with rat peritoneal macrophages. Internalized hPEMC seen in the images had irregular deformed shapes, a phenomenon observed in previous studies [4,27]. This most likely, from compression of the multilayered shells of the capsules during their uptake by the cells, as well as the action of the acidified milieu created inside the phagosomal compartments upon fusion with the lysosomes during intracellular trafficking [33]. The appearance of fluorescent probes in the cytoplasm indicates the successful delivery of the model compounds from the capsules to the cells.

As a result, we have been able to produce stable preparations of unaggregated hPEMC which were efficiently internalized by live cells with subsequent release of combined hydrophilic and hydrophobic cargo, without exerting any apparent cytotoxic side effects. The polyelectrolyte systems developed in our study could potentially find medical application as carriers for therapeutic compounds with low bioavailability and/or high toxicity, as well as antibodies, cytokines, and other hydrophilic bioactive macromolecules. This approach will enable synergistic action of encapsulated compounds to enhance the therapeutic effect.

## 5. Conclusions

In this study, we demonstrate successful production of hybrid microcapsules by employing the method of layer-by-layer electrostatic assembly in combination with degradable biomineral particles of mixed CaCO_3_-protein composition. Ultrastructural analysis of the resulting microcapsules revealed the absence of regular structure in their external shell, which can be regarded as an advantage since such architecture accelerates hPEMC degradation in vitro. Integration of magnetic particles into hPEMC for the purposes of electron microscopy imaging may find further development for practical applications, since this type of modification can be utilized to control the delivery of hPEMC to the target tissues/organs, resulting in significantly improved therapeutic effect; for creating targeted therapeutic hyperthermia; as well as for controlled drug release, by incorporating thermo-sensitive polymers into the capsules. Additionally, the presence of magnetite nanoparticles in microcapsules makes them a potential instrument for MR imaging.

We have demonstrated highly efficient simultaneous loading of PEMC with a combination of hydrophilic and hydrophobic compounds. All the variations of the hPEMC design tested in our study have demonstrated high capacity of protein encapsulation inside internal cavity of the carrier (86.0 ± 0.88–89.93 ± 0.33%), as well as efficient incorporation of the RBITC model compound into the hydrophobic layer, with encapsulation efficiency of 77% and higher. The microcapsules demonstrated high storage stability, regardless of the composition and thickness of their polyelectrolyte shell. Analysis of biocompatibility of hPEMC produced by us showed absence of toxicity toward the tested stable cell lines and rat primary macrophages. We can conclude that our hPEMC represent a multifunctional platform with the capability for simultaneous release of combinations of therapeutic compounds with diverse physico-chemical properties; this platform can be widely applicable as a co-delivery system for a wide array of anti-inflammatory and antitumor drugs.

## Figures and Tables

**Figure 1 polymers-14-00931-f001:**
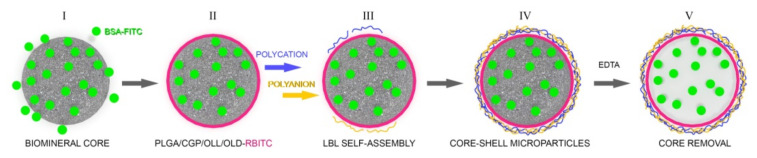
Schematic illustration showing the preparation of hybrid microcapsules. I. Formation of a composite CaCO_3_ microspherulite containing the model hydrophilic compound (BSA-FITC) by coprecipitation. II. Coating the composite microspherulite with a hydrophobic layer of PLGA (CGP, OLL, OLD) containing the model hydrophobic compound (RBITC). III–IV. Sequential adsorption of oppositely charged polyelectrolytes on the surface of the microspherulite. V. Demineralization of the carbonate matrix in 0.2 M EDTA.

**Figure 2 polymers-14-00931-f002:**
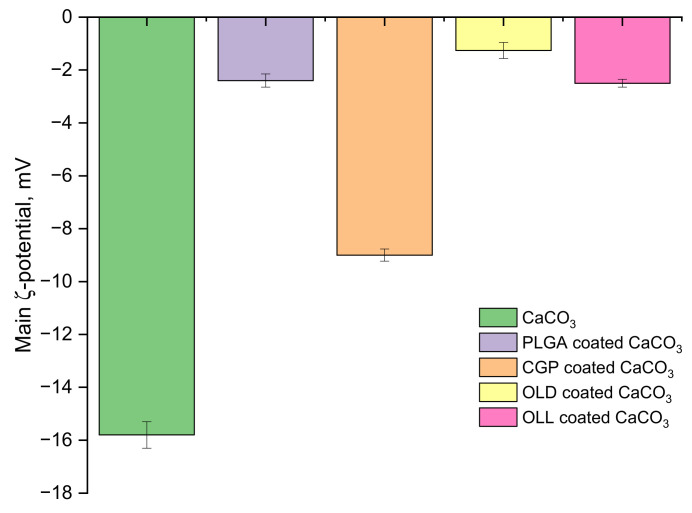
ζ-potential values for CaCO_3_ without coating, PLGA-coated CaCO_3_ templates, CGP-coated CaCO_3_ templates, OLL-coated CaCO_3_ templates, and OLD-coated CaCO_3_ templates.

**Figure 3 polymers-14-00931-f003:**
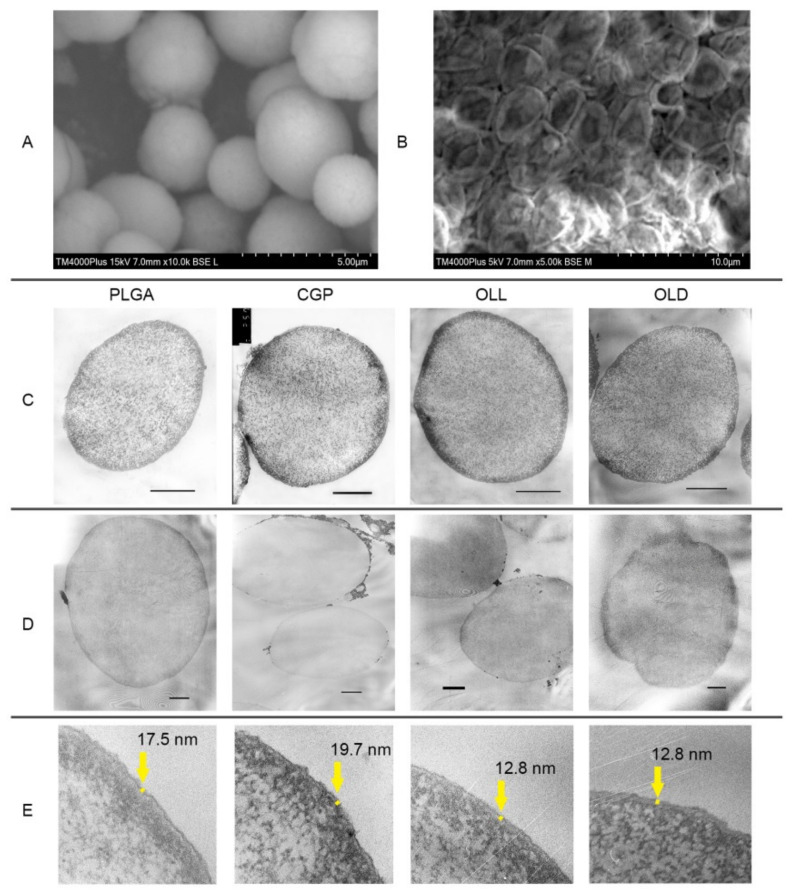
Transmission electron micrographs of ultrathin slices of microcapsules with the composition PLGA/CGP/OLL/OLD(PAr/DS)_2_. (**A**) microparticles CaCO_3_, (**B**) hybrid microcapsules after drying, (**C**) contrasting by sodium uranyl acetate and lead citrate; scale bar 3 µm. (**D**) without contrasting; scale bar 1 µm. (**E**) thickness of the layer hybrid microcapsules.

**Figure 4 polymers-14-00931-f004:**
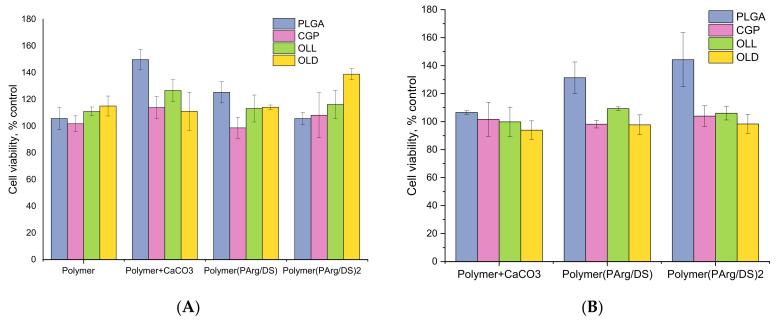
The effect of hPEMC and their structural components on the viability of mouse L929 fibroblasts. (**A**) capsule-to-cell ratio 2:1; (**B**) capsule-to-cell ratio 10:1. Cell viability indicated as % of control; each point is a mean of 3 independent experiments. 5 × 10^3^ cells per of a 96-well plate; polymer concentration (PLGA, CGP, OLL, OLD)—0.5 µg/mL; exposure time—72 h.

**Figure 5 polymers-14-00931-f005:**
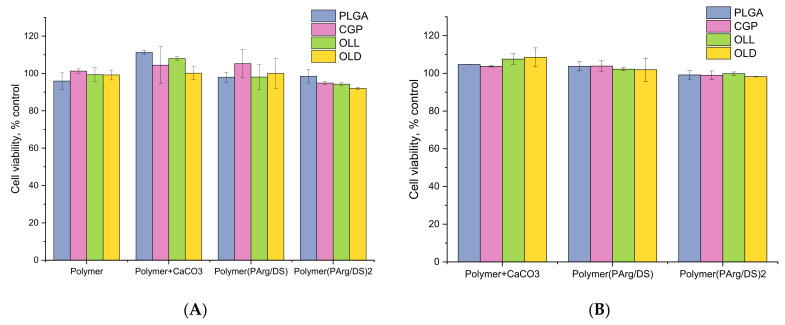
The effect of hPEMC and their structural components on the viability of human promyelocytic HL-60 cells. (**A**) capsule-to-cell ratio 2:1; (**B**) capsule-to-cell ratio 10:1. Cell viability indicated as % of control; each point is a mean of 3 independent experiments. 5 × 10^3^ cells per of a 96-well plate; polymer concentration (PLGA, CGP, OLL, OLD)—0.5 µg/mL; exposure time—72 h.

**Figure 6 polymers-14-00931-f006:**
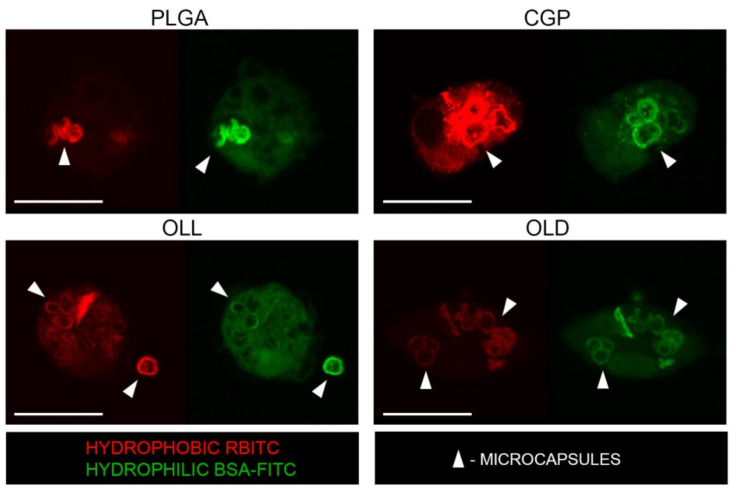
Reprepsentative confocal laser scanning microscope images showing internalization of hPEMC by rat macrophages after 48 h of exposure.

**Table 1 polymers-14-00931-t001:** The efficiency of encapsulation of model compounds (BSA-FITC and RBITC) into the cavity and the hydrophobic layer of hPEMC with different composition.

Type of hPEMC	Encapsulation Efficiency Model Compound BSA-FITC into Cavity hPEMC, %	Encapsulation Efficiency Model Compound RBITC into Hydrophobic Layer, %
PLGA (PAH/PSS)	89.04 ± 0.65	77.46 ± 2.34
CGP (PAH/PSS)	86.20 ± 0.88	85.16 ± 0.86
OLL (PAH/PSS)	88.64 ± 0.98	86.71 ± 2.76
OLD (PAH/PSS)	89.05 ± 0.59	86.60 ± 5.68
PLGA (PAH/PSS)_2_	89.93 ± 0.33	78.96 ± 4.65
CGP (PAH/PSS)_2_	88.94 ± 2.31	91.10 ± 3.62
OLL (PAH/PSS)_2_	89.83 ± 5.81	90.79 ± 2.19
OLD (PAH/PSS)_2_	89.86 ± 0.57	86.66 ± 1.67

**Table 2 polymers-14-00931-t002:** Cumulative release of model compounds from hPEMC. (480 h incubation, each experiment performed in triplicate and average values indicated).

Type of hPEMC	Cumulative Release, BSA-FITC, %		Cumulative Release, RBITC, %	
	4 °C	37 °C	4 °C	37 °C
PLGA (PAH/PSS)	1.09 ± 0.01	1.93 ± 0.28	0.62 ± 0.01	0.46 ± 0.08
CGP (PAH/PSS)	0.75 ± 0.10	2.10 ± 0.08	1.32 ± 0.01	0.61 ± 0.02
OLL (PAH/PSS)	1.30 ± 0.15	3.29 ± 0.014	0.89 ± 0.20	1.69 ± 0.25
OLD (PAH/PSS)	0.89 ± 0.35	2.29 ± 0.88	0.63 ± 0.34	0.78 ± 0.01
PLGA (PAH/PSS)_2_	0.80 ± 0.05	3.29 ± 0.90	0.92 ± 0.09	2.19 ± 0.99
CGP (PAH/PSS)_2_	0.53 ± 0.01	2.31 ± 0.35	0.66 ± 0.21	1.43 ± 0.58
OLL (PAH/PSS)_2_	1.05 ± 0.17	2.67 ± 0.55	0.66 ± 0.03	1.39 ± 0.71
OLD (PAH/PSS)_2_	0.77 ± 0.11	2.33 ± 1.40	0.47 ± 0.08	1.28 ± 0.84

## Data Availability

Data sharing not applicable.

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
