# Peer review of "Universal Microcarriers Based on Natural and Synthetic Polymers for Co-Delivery of Hydrophilic and Hydrophobic Compounds"

_polymers, 2022, doi:10.3390/polym14050931_

Round 1
Reviewer 1 Report
Dear Authors
The submitted work is interesting for the readers and addressing a point of need in the development of drug delivery system.
The following comments and suggessions have to be considered by your side during the revision before your work consider for publication.
Comments
Title:
The title should contains an indication of the polymers used to develope the drug delivery matrices.
It is better to replace the word"containers" by "Carriers"
Abstract
The word "Polyelectorlite" should be correcetd.
The abstarct must mentioned the full name of the mentionned abbreviation when appear for the first time.
The type of the polymers used in the study should be stated.
Introduction
The word "Polyelectorlite" should be correcetd.
It is better to replace the word"containers" by "Carriers".
The authors should cited a reference at the end of the sentence "Hybrid microcapsules were produced by a modified method of sequential ad-71 sorption of oppositely charged polyelectrolytes on the surface of layer-by-layer (LbL) 72 carbonate matrix with modifications [Ref].
Materials and Methods
The synthesis of chitosan/gelatin/polylaactide copolymer (CGP) should be mentioned under section 2.1.
The text in section 2.1. is very confusing and hard to follow by the readers. Please consider re-writting.
Author Response
См вложение

Reviewer 2 Report
In this manuscript, the authors introduce a novel drug delivery system capable of combination drug therapy, using a hybrid polyelectrolyte microcapsules. The paper focuses on the characterization of the microcapsules. The content may be of interest to the readers of the journal Polymers, but some points should be considered for publication.
- In the introduction, the authors should describe the significance of the novel system more thoroughly. For example, the authors show prevention of premature release as one of the merits through the results and discussion. However, such hPEMC's significance, in the context of other delivery systems that fall short of this need, should be mentioned in the introduction.
- The method of CGP synthesis should be at least briefly described, and not just referenced.
- What is the actual loading capacity of BSA-FITC and RBITC? High loading efficiency does not always correlate with high loading capacity.
- There should be further discussion of the potential use of hPEMC system, in the context of the treatment dose. Its slow release, and depending on what the loading capacity is, may require large dose of hPEMC to be injected to provide therapeutic effective concentration of the combo drugs. How does this relate to the in vitro cytotoxicity experiment in terms of the different ratios of hPEMC to cells tested? How would this affect a potential in vivo experiment?
- In figure 6, there is a typo. Hidrophobic should be hydrophobic, and hidrophilic should be hydrophilic.
- In figure 6, why is BSA-FITC signal in the periphery of hPEMC (like a ring)? Shouldn't BSA-FITC be loaded in the center, and thus should appear as a solid round signal?
- Cellular uptake experiment can be performed with flow cytometry to yield a better quantitative data than a qualitative description of "3 to 4 microcapsules were phagocytosed."
- It would be great to see a discussion on the key differences between different hPEMCs synthesized and characterized, and for which applications one might be more advantageous than others.
Reviewer 3 Report
Please see the attachment for the comments.
